# Dynamic three-sided matching model for personnel–robot-position matching problem in intelligent environments

**Zhi-chao Liang**[1], **Yu Yang**[1], **Qiu Xie**[2]*, **Jing Wang**[1], **Xue-jiao Zhang**[1], **Bao-dong Li**[3]

**1** College of Mechanical and Vehicle Engineering, Chongqing University, Chongqing, China, **2** Research Center for Construction Economics and Management, School of Management Science and Real Estate, Chongqing University, Chongqing, China, **3** School of Economics and Business Administration, Chongqing University of Education, Chongqing, China

* xieqiu@cqu.edu.cn

**Data Availability Statement:** All relevant data are within the paper and its Supporting Information files.

**Funding:** This study was supported by Fundamental Research Funds for the Central

## Abstract

In recent years, intelligent robots have facilitated intelligent production, and a new type of problem (personnel–robot-position matching (PRPM)) has been encountered in personnel–position matching (PPM). In this study, a dynamic three-sided matching model is proposed to solve the PRPM problem in an intelligent production line based on man–machine collaboration. The first issue considered is setting the dynamic reference point, which is addressed in the information evaluation phase by proposing a method for setting the dynamic reference point based on the prospect theory. Another important issue involves multistage preference information integration, wherein a probability density function and a value function are introduced. Considering the attenuation of preference information in a time series, the attenuation index model is introduced to calculate the satisfaction matrix. Furthermore, a dynamic three-sided matching model is established. Additionally, a multi-objective decision-making model is established to optimize the matching of multiple sides (personnel, intelligent robots, and positions). Subsequently, the model is transformed into a single objective model using the triangular balance principle, which is introduced to obtain the final optimisation results in this modelling process. A case study is presented to illustrate the practicality of the dynamic three-sided matching model in intelligent environments. The results indicate that this model can solve the PRPM problem in an intelligent production line.

## 1. Introduction

Personnel–position matching (PPM) decision-making is significant for enterprise management [1, 2]. Several models and methods have been proposed to further improve the efficiency of management decision-making, such as the analytic hierarchy process (AHP) [3, 4] and artificial neural networks [5]. With the development of artificial intelligence technology, the conventional manufacturing environment has become intelligent [6–8]. In a weak-intelligence environment, personnel and machine are replaceable; thus, the PPM problem is transformed into a PMPM problem [9]. The interval-valued intuitionistic fuzzy two-sided matching model

Universities-China (No. 2020CDJSK03PT08 and 2021CDJSKJC17) and the Scientific and Technological Research Program of Chongqing Municipal Education Commission (Grant No. KJQN202101617). The funders (No. 2020CDJSK03PT08, 2021CDJSKJC17) provide the publication fee for our study. The funder (No. KJQN202101617) provides consulting fee for our study.

**Competing interests:** The authors have declared that no competing interests exist.

is proposed to solve this problem. The two-sided matching theory has been applied in different fields, for example, the problem of matching between knowledge suppliers and knowledge demanders [10] and matching the supply and demand of green building technology [11]. However, the role of personnel in a strong intelligent manufacturing environment has transformed into that of supervisors or auxiliary personnel. In this case, a new problem termed as personnel-robot-position matching (PRPM) is derived from PPM. This problem differs from PMPM. The two-sided matching strategy does not apply to PRPM. Therefore, it is necessary to determine an appropriate method for solving the PRPM problem in intelligent environments.

Meanwhile, a reasonable solution for PRPM is the key to realising human–machine collaboration in an advanced industry wherein the combination of human and robot skills can increase the flexibility of manufacturing systems [12]. The main concept of human–machine collaboration involves combining human capabilities with robot capabilities. Humans possess natural flexibility, intelligence, flexibility, and the capability to solve problems, whereas robots provide the advantages of accuracy, strength, and repeatability in task performance. Therefore, it is necessary to focus on safety [13], task allocation [14], cognitive load, ergonomics, acceptability, and other factors involved in personnel management [15]. With respect to PRPM, Zacharias et al. examined the collaborative method of personnel and intelligent robot arm from the perspective of ergonomics [16]. Yin et al. proposed a man–machine configuration design method based on the imagination thinking from the perspective of human–machine interaction [17]. Zhang et al. proposed a man–machine collaborative simulation model to configure the typical production units of key parts in the motorcycle engine production line considering skill type, skill level, and man–machine division of labour [18]. Considering the skill level of personnel and complexity of machine operation, Liu and Yang adopted the concept of a non-dominated solution set of a genetic algorithm to solve the human–machine configuration problem of the production line with the objectives of resource utilisation, operation cost, and delivery time [19]. We observe that these studies focused on a microscopic level and did not address the entire manufacturing system. Another part of these studies considered the skill level of personnel from the perspective of the entire manufacturing system. However, in these studies, factors such as safety and acceptability and the integration of the human-oriented manufacturing concept were not considered.

Further literature review revealed that the key issue of PRPM is the allocation of production resources (personnel and intelligent robots) in the production system. Therefore, PRPM is regarded as a three-sided matching problem. Researchers at the Yanshan University examined the matching of three types of functional personnel (business, design, and production) in the project development team considering the preference order and probability hesitation fuzzy element [20, 21]. Additionally, Yuan examined the matching of supply, transportation, and demand in the coal market under the seven granularity language evaluation set [22]. Yi designed the grey correlation degree calculation method based on the cloud model and applied it to the three-sided matching problem of 'students–in school tutors–out of school tutors' in industry–university research collaboration [23]. The three-sided matching theory has also been applied in the medical environment: three-sided kidney transplantation [24] and three-sided collaboration among doctors, anaesthesiologists, and nurses [25]. Fuzzy sets have been widely used in the three-sided matching problem. The fuzzy set theory has also been successfully applied in the big medical data decision-making intelligent system to observe the curative powers of the deterministic treatment method on patients in real time [26]. However, in the matching decision-making process, the single-stage preference information cannot fully reflect the preference degree of the matching subject. Therefore, most manufacturing enterprises use multi-stage evaluation. In the existing studies, the three-sided matching method continues to be static, thereby avoiding the dynamic characteristic of multi-stage preference information.

In the multi-stage dynamic matching process, the preference information of different time dimensions is determined based on the decision-making environment. There are complex psychological factors in the preference information provided by the decision-makers. Decision maker display risk aversion awareness when they encounter benefits and risk preference tendency when they encounter losses. Moreover, the losses are more sensitive than the benefits. Therefore, it is necessary to completely consider the psychological factors of the preference information provided by the decision-makers to scientifically and reasonably reflect the degree of preference. Furthermore, certain studies proposed an incomplete fuzzy preference relationship based on the disappointment theory to represent the comparative preference information of other matching objects. The aim involves describing the disappointment and elation among the matching objects [27]. However, the prospect theory considers that decision-makers should set reference points according to their degree of preference. The setting of multi-stage dynamic reference points can reflect horizontal differences between matching subjects and consider the variations in different stages. Furthermore, decision-makers possess different information in different time series. In general, closer to the end of the decision-making process, the information that the matching subject has is more complete, and the impact on the matching result is higher. Conversely, in the early stage of decision-making, the impact of the information that the matching subject possesses on the final matching result is low. This method comprehensively considers the psychological factors of decision-makers and reflects the impact of preference information in different stages on the final decision-making results. Therefore, in this study, the prospect theory and the attenuation index model are introduced to integrate the multi-stage preference information.

The aforementioned studies are key to the solution of the PRPM problem. This study was motivated by this background. In this study, a dynamic three-sided matching decision-making method is proposed for the dynamic three-sided matching problem of personnel-robot position in an intelligent assembly production line. The key to dynamic three-sided matching is the application of the prospect theory.

The main contributions of this study are as follows:

1. A scientific and systematic man–machine collaborative fuzzy configuration method is established to solve the problem of man–machine collaborative configuration in a flexible assembly manufacturing system.

2. The proposed method considers the psychology of decision-makers and variation in the trend of the preference information of three-sided matching subjects in different stages, and it realises multi-stage dynamic decision-making.

In Section 2, certain fundamental concepts with respect to the two-sided matching problem are presented. Section 3 describes the aggregation of the multi-stage preference information. The specific steps include the following: setting of dynamic reference points, calculation of the income–loss matrix, determination of the prospect value matrix, and calculation of the satisfaction matrix. Section 4 describes the dynamic three-sided matching model. In Section 5, its solving strategy is presented. In Section 6, a case study for verifying the effectiveness of the model is presented, and the model is discussed. In Section 7, the conclusions of this study are presented.

## 2. Dynamic three-sided matching problem

In an intelligent assembly line, intelligent robots and personnel collaborate to complete the production tasks in a production position and realise optimal configuration. Therefore, the problem is defined as three-sided matching. There are three sets in an intelligent assembly

line: personnel set $A$, intelligent robot set $B$, and position set $C$. Let $A = \{a_1, a_2, \cdots a_m\}$. Here, $m \in N$, $a_i$ denotes the $i$th personnel, and $i = 1, 2 \cdots n$. Let $B = \{b_1, b_2, \cdots b_n\}$. Here, $n \in N$, $b_j$ denotes the $j$th intelligent robot, and $j = 1, 2 \cdots n$. Let $C = \{c_1, c_2, \cdots c_l\}$. Here, $l \in N$, $c_k$ represents the $k$th position, and $k = 1, 2 \cdots l$. The quantitative relationship among personnel, intelligent robots, and positions is $m \geq n \geq l \geq 2$.

**Definition 1** [28]: In the three-sided matching model, the preferred directed edge between different subjects is termed as a directed preference. If the subject of side $A$ prefers the subject of side $B$, then the preferred edge $\overrightarrow{AB}$ is formed. If the subject of side $B$ prefers the subject of side $A$, then the preferred edge $\overrightarrow{BA}$ is formed. In the three-sided matching problem, there is at least one preferred edge on either side.

**Definition 2** [29]: In the three-sided matching model, if the subjects on either side exhibit only one directed preference edge, the problem is termed as unidirectional matching. In three-sided unidirectional matching, if the preferences between three-sided subjects form a cyclic relationship, then the problem is termed as three-sided cyclic matching. Otherwise, it is termed as three-sided unidirectional acyclic matching.

The PRPM problem in an intelligent assembly line is similar to the collaborative relationship between surgeons, anaesthesiologists, and nurses [25]. Therefore, the preference between the three-sided subjects of personnel–robot positions is a three-sided one-way acyclic relationship. In the personnel–robots position three-sided matching system, a directed group $(a_i, b_j, c_k) \in A \times B \times C$ is composed of three sides. Here, $i = 1, 2 \cdots m$, $j = 1, 2 \cdots n$, and $k = 1, 2 \cdots l$. A set composed of all matching groups is termed as a match of the matching system. It is set as $M$. A set $B$ composed of all matching groups is shown in Fig 1.

In the model, three lines denote the matching relationship between different subjects: personnel have a directed preference for the intelligent robot, personnel have a directed preference for the position, and the intelligent robot has a direct preference for the position. Therefore, the PRPM problem in an intelligent assembly line is identical to a three-sided unidirectional acyclic problem. Let $N(M, x)$ indicate the number of matching groups. Here, $N(M, x) = 0$ $or$ 1, $x \in A \cup B \cup C$. $N(M, x) = 0$ indicates that it is idle and that no entity is matched in the matching system. Furthermore, $N(M, x) = 1$ indicates that an entity is matched in the matching system.

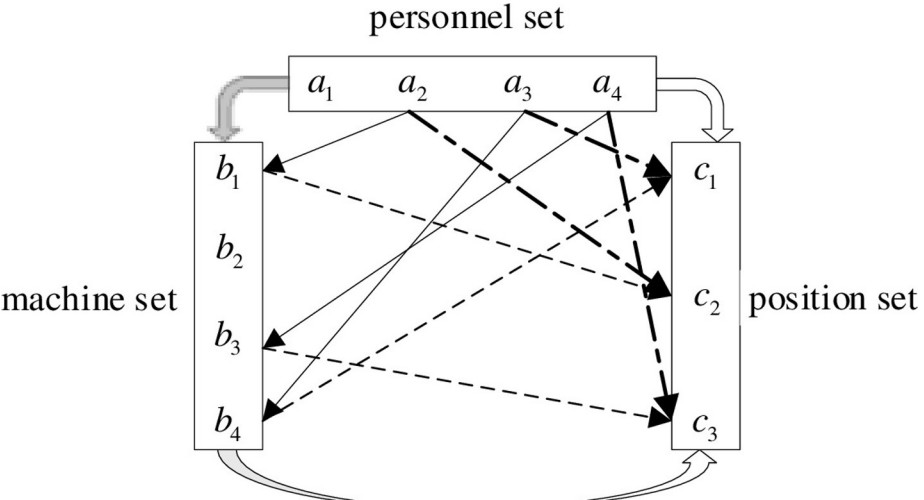

**Fig 1. Three-side matching model of personnel–robot position.**

In this matching system, the preference information between subjects is obtained through expert multi-stage evaluation. Set $T = \{t_1, t_2, \cdots t_p\}$ is used to represent the set of evaluation stages. The preference degree of personnel $a_i$ for intelligent robot $b_j$ in stage $t_k$ is expressed as $\vec{d_{ij}^{ab}}(t_k)$, that of personnel $a_i$ for position $c_k$ in stage $t_k$ is expressed as $\vec{d_{ik}^{ac}}(t_k)$, and that of intelligent robot $b_j$ for position $c_k$ in stage $t_k$ is expressed as $\vec{d_{jk}^{bc}}(t_k)$. Similarly, the preference information matrix between different stages can be obtained. The preference information matrix of personnel to the intelligent robot in a stage is expressed as $D^{\vec{ab}}(t_k)$, that of personnel to position in a stage is expressed as $D^{\vec{ac}}(t_k)$, and that of intelligent robot to position in a stage is expressed as $D^{\vec{bc}}(t_k)$. The problem investigated in this study corresponds to the identification of a method to transform the multi-stage evaluation information of a three-sided matching subject into a reasonable configuration scheme.

## 3. Multi-stage preference information aggregation

### 3.1 Obtaining interval numbers based on uncertain linguistic preference information

In management decision-making, owing to the complexity of the environment and limitations of cognition [30], the evaluation information can be presented in the form of fuzzy language. For example, words such as 'excellent', 'good', 'medium', and 'poor' are used to describe the characteristics of objective entities. Given that the preference information in the form of language is more in line with the fuzziness and uncertainty of the human thought process, it aids decision-makers in identifying the real state of things and reduce their cognitive pressure [31]. In recent years, the study of uncertain language preferences has also attracted the attention of researchers [32]. Additionally, the digital transformation of fuzzy language is an important step in the decision-making process. For this problem, Xu designed the fuzzy language evaluation scale method [33]. The mapping between fuzzy language and the corresponding interval number is as follows: excellent, [0.8,1]; good, [0.6,0.8]; medium, [0.4,0.6]; poor, [0.2,0.4]; special poor, [0,0.2]. Thus, in this study, the uncertain linguistic preference information between three-sided subjects is quantified via the fuzzy language evaluation scale method (see Table 1).

### 3.2 Setting dynamic reference point

The setting of the dynamic reference point is the key step in the process of dynamic three-sided matching. The concept of reference point originates from the prospect theory. The decision-maker provides information of degree of income or loss of the evaluation object based on the reference point. This method can maintain the consistency of the evaluation. The setting of reference points is generally based on the mean method or zero-point method. In this study, the mean method is used to set the initial reference points of each stage. A more reasonable degree of preference is reflected by a horizontal comparison of internal data in the same stage. However, in the multi-stage dynamic matching process, the preference information of different time dimensions is determined based on the environment at the time of decision-making. Therefore, integration of the preference information of different time dimensions is a technical problem that should be solved urgently.

**Table 1. Evaluation scale of inaccurate linguistic preference information.**

| Not match | Fairly match | Quite match | Rather match | Very match |
|---|---|---|---|---|
| [0,0.2] | [0.2,0.4] | [0.4,0.6] | [0.6,0.8] | [0.8,1] |

Accordingly, a multi-stage dynamic reference point setting method based on overall preference expectation is proposed. First, the average method is used to calculate the initial reference point of each stage. Then, the overall preference expectation is calculated based on the initial reference point of each stage. Finally, the dynamic reference point of each stage is calculated with reference to the overall preference expectation. The initial reference point is defined as follows.

**Definition 3**: Set the initial reference point of a certain stage as $\bar{d}(t_k)$, indicating the expected value in the $t_k$ stage. In the reference matrix $D^{\vec{ab}}(t_k)$ of personnel to the intelligent robot, $\bar{d}^{\vec{ab}}(t_k)$ denotes the initial reference point of personnel to the intelligent robot at stage $t_k$. The calculation formula is as follows.

$$\bar{d}^{\vec{ab}}(t_k) = \frac{1}{m \times n} \sum_{i=1}^{m} \sum_{j=1}^{n} D^{\vec{ab}}(t_k) \tag{1}$$

Similarly, the preference information matrix $D^{\vec{ac}}(t_k)$ of personnel to position, $\bar{d}^{\vec{ac}}(t_k)$, denotes the initial reference point of personnel to the position at stage $t_k$. In the preference information matrix $D^{\vec{bc}}(t_k)$ of intelligent robot to the position, $\bar{d}^{\vec{bc}}(t_k)$ denotes the initial reference point of the preference of intelligent robot to the position at stage $t_k$. The calculation formulae are as follows:

$$\bar{d}^{\vec{ac}}(t_k) = \frac{1}{m \times l} \sum_{i=1}^{m} \sum_{k=1}^{l} D^{\vec{ac}}(t_k) \tag{2}$$

$$\bar{d}^{\vec{bc}}(t_k) = \frac{1}{n \times l} \sum_{j=1}^{n} \sum_{k=1}^{l} D^{\vec{bc}}(t_k) \tag{3}$$

where $\bar{d}^{\vec{ab}}$, $\bar{d}^{\vec{ac}}$, and $\bar{d}^{\vec{bc}}$ denote the overall preference and expectation of personnel to the intelligent robot, personnel to the position, and intelligent robot to the position, respectively. The calculation formulae are as follows:

$$\bar{d}^{\vec{ab}} = \frac{1}{p} \sum_{k=1}^{p} \bar{d}^{\vec{ab}}(t_k) \tag{4}$$

$$\bar{d}^{\vec{ac}} = \frac{1}{p} \sum_{k=1}^{p} \bar{d}^{\vec{ac}}(t_k) \tag{5}$$

$$\bar{d}^{\vec{bc}} = \frac{1}{p} \sum_{k=1}^{p} \bar{d}^{\vec{bc}}(t_k) \tag{6}$$

The dynamic reference points of each stage are calculated based on the initial reference point and overall preference expectation. The dynamic reference point of personnel to the intelligent robot is set as $r^{\vec{ab}}(t_k)$, that of personnel to position is set as $r^{\vec{ac}}(t_k)$, and that of the

**Table 2. Six quantitative relationships.**

| case | Quantitative relationship of interval values |
|------|----------------------------------------------|
| Case 1 | $d_{ij}^{\vec{ab}}(t_k)^- < d_{ij}^{\vec{ab}}(t_k)^+ < r^{\vec{ab}}(t_k)^- < r^{\vec{ab}}(t_k)^+$ |
| Case 2 | $r^{\vec{ab}}(t_k)^- < r^{\vec{ab}}(t_k)^+ < d_{ij}^{\vec{ab}}(t_k)^- < d_{ij}^{\vec{ab}}(t_k)^+$ |
| Case 3 | $d_{ij}^{\vec{ab}}(t_k)^- < r^{\vec{ab}}(t_k)^- < d_{ij}^{\vec{ab}}(t_k)^+ < r^{\vec{ab}}(t_k)^+$ |
| Case 4 | $r^{\vec{ab}}(t_k)^- < d_{ij}^{\vec{ab}}(t_k)^- < r^{\vec{ab}}(t_k)^+ < d_{ij}^{\vec{ab}}(t_k)^+$ |
| Case 5 | $r^{\vec{ab}}(t_k)^- < d_{ij}^{\vec{ab}}(t_k)^- < d_{ij}^{\vec{ab}}(t_k)^+ < r^{\vec{ab}}(t_k)^+$ |
| Case 6 | $d_{ij}^{\vec{ab}}(t_k)^- < r^{\vec{ab}}(t_k)^- < r^{\vec{ab}}(t_k)^+ < d_{ij}^{\vec{ab}}(t_k)^+$ |

intelligent robot to position is set as $r^{\vec{bc}}(t_k)$. The calculation formulae are as follows:

$$r^{\vec{ab}}(t_k) = \frac{1}{2}(\bar{d}^{\vec{ab}}(t_k) + \bar{d}^{\vec{ab}}) \tag{7}$$

$$r^{\vec{ac}}(t_k) = \frac{1}{2}(\bar{d}^{\vec{ac}}(t_k) + \bar{d}^{\vec{ac}}) \tag{8}$$

$$r^{\vec{bc}}(t_k) = \frac{1}{2}(\bar{d}^{\vec{bc}}(t_k) + \bar{d}^{\vec{bc}}) \tag{9}$$

The core of the dynamic reference point setting method proposed in this study involves integrating the reference point of each stage with the overall expectation. The advantage is that it can prevent inaccurate evaluation due to the incomplete information in the evaluation process at a certain stage. The following work is performed to calculate the prospect value of each stage based on the obtained dynamic reference points.

## 3.3 Calculation of profit and loss based on the probability density function

The main aim of this section involves calculating the return value and loss value based on the dynamic reference point via the probability density function of the interval number. There are six positional relationships between the evaluation value and dynamic reference point [34]. Given the involvement of many parameters, in this study, the calculation process of the prospect value of the intelligent robot is considered as an example.

**Definition 4:** In a certain interval number $Z = [Z^-, Z^+]$, set $x$ denotes any value on the interval number. Then, the probability density function of $x$ is as follows:

$$f(x) = \frac{1}{Z^+ - Z^-} \tag{10}$$

where $Z^- \leq x \leq Z^+$, $\int_{Z^-}^{Z^+} f(x)\, dx = 1$. For any $x$, $f(x) \geq 0$.

In the multi-stage evaluation process of an intelligent robot, the dynamic reference point and preference information exhibit the characteristics of an interval number. Therefore, the relationship between them can be classified into the following six cases (see Table 2).

The quantitative relationship in the first case is $d_{ij}^{\vec{ab}}(t_k)^- d_{ij}^{\vec{ab}}(t_k)^+ < r^{\vec{ab}}(t_k)^- < r^{\vec{ab}}(t_k)^+$. The preference information is significantly smaller than the dynamic reference point. According to the prospect theory, it can be conjectured that there is no gain in the terms of personnel for the intelligent robot. The data in any case includes the lower limit of dynamic reference points, the

upper limit of dynamic reference points, the lower limit of preference information, and the upper limit of preference information. The corresponding four types of data can have six different distribution forms to represent the relationship between preference information and dynamic reference points. Therefore, it can be expressed as follows:

$$G_{ij}^{\vec{ab}}(t_k) = 0 \tag{11}$$

According to definition 4, the loss in the terms of personnel for the intelligent robot is as follows:

$$L_{ij}^{\vec{ab}}(t_k) = \int_{d_{ij}^{\vec{ab}}(t_k)^-}^{d_{ij}^{\vec{ab}}(t_k)^+} (x - r^{\vec{ab}}(t_k)^-)f(x)dx = 0.5\,(d_{ij}^{\vec{ab}}(t_k)^- + d_{ij}^{\vec{ab}}(t_k)^+) - r^{\vec{ab}}(t_k)^- \tag{12}$$

The quantitative relationship in the second case is $r^{\vec{ab}}(t_k)^- < r^{\vec{ab}}(t_k)^+ < d_{ij}^{\vec{ab}}(t_k)^- < d_{ij}^{\vec{ab}}(t_k)^+$. The preference information is significantly higher than the dynamic reference point. According to the prospect theory, it can be conjectured that there are benefits for personnel relative to the intelligent robot. However, the loss value is zero. Therefore, it can be expressed as follows:

$$G_{ij}^{\vec{ab}}(t_k) = \int_{d_{ij}^{\vec{ab}}(t_k)^-}^{d_{ij}^{\vec{ab}}(t_k)^+} (x - r^{\vec{ab}}(t_k)^-)f(x)dx = 0.5\,(d_{ij}^{\vec{ab}}(t_k)^- + d_{ij}^{\vec{ab}}(t_k)^+) - r^{\vec{ab}}(t_k)^+ \tag{13}$$

$$L_{ij}^{\vec{ab}}(t_k) = 0 \tag{14}$$

The quantitative relationship in the third case is $d_{ij}^{\vec{ab}}(t_k)^- < r^{\vec{ab}}(t_k)^- < d_{ij}^{\vec{ab}}(t_k)^+ < r^{\vec{ab}}(t_k)^+$. There is an intersection between the preference information and dynamic reference point. The lower limit of the dynamic reference point is lower than the upper limit of the preference information interval value. However, it is higher than the lower limit of the preference information interval value. Herein, the upper limit of the dynamic reference point is the highest. Therefore, according to the prospect theory, it can be conjectured that there are certain benefits and certain losses for personnel to the intelligent robot. It should be noted that part $\left[ r^{\vec{ab}}(t_k)^-, d_{ij}^{\vec{ab}}(t_k)^+ \right]$ incurs no gain or loss in terms of personnel for the intelligent robot when $\left[ r^{\vec{ab}}(t_k)^-, d_{ij}^{\vec{ab}}(t_k)^+ \right] \in \left[ r^{\vec{ab}}(t_k)^-, r^{\vec{ab}}(t_k)^+ \right]$. Therefore, it is necessary to consider that another part $\left[ d_{ij}^{\vec{ab}}(t_k)^-, r^{\vec{ab}}(t_k)^- \right]$ is less than $\left[ d_{ij}^{\vec{ab}}(t_k)^+, r^{\vec{ab}}(t_k)^+ \right]$. In this case, there is a loss value for personnel of the intelligent robot. However, the benefit value is zero. Therefore, they are expressed as follows.

$$G_{ij}^{\vec{ab}}(t_k) = 0 \tag{15}$$

$$L_{ij}^{\vec{ab}}(t_k) = \int_{d_{ij}^{\vec{ab}}(t_k)^-}^{r^{\vec{ab}}(t_k)^-} (x - r^{\vec{ab}}(t_k)^-)f(x)dx = 0.5\,\left( d_{ij}^{\vec{ab}}(t_k)^- - r^{\vec{ab}}(t_k)^- \right) \tag{16}$$

In the third case, if $r^{\vec{ab}}(t_k)^- = d_{ij}^{\vec{ab}}(t_k)^+$, then intersection $\left[ r^{\vec{ab}}(t_k)^-, d_{ij}^{\vec{ab}}(t_k)^+ \right]$ is a reference point. It evolves into the first case. However, Eq (13) continues to hold.

In the fourth case, $r^{\vec{ab}}(t_k)^- < d_{ij}^{\vec{ab}}(t_k)^- < r^{\vec{ab}}(t_k)^+ < d_{ij}^{\vec{ab}}(t_k)^+$. It is similar to the third case. The preference information and dynamic reference points also intersect. Similarly, the profit value and loss value of personnel to the intelligent robot can be obtained as follows:

$$G_{ij}^{\vec{ab}}(t_k) = \int_{r^{\vec{ab}}(t_k)^+}^{d_{ij}^{\vec{ab}}(t_k)^+} (x - r^{\vec{ab}}(t_k)^+)f(x)dx = 0.5\left(d_{ij}^{\vec{ab}}(t_k)^+ - r^{\vec{ab}}(t_k)^+\right) \tag{17}$$

$$L_{ij}^{\vec{ab}}(t_k) = 0 \tag{18}$$

In the fifth case, $r^{\vec{ab}}(t_k)^- < d_{ij}^{\vec{ab}}(t_k)^- < d_{ij}^{\vec{ab}}(t_k)^+ < r^{\vec{ab}}(t_k)^+$. An inclusion relationship exists between the preference information and dynamic reference points. The interval number of preference information is a subset of the interval number of dynamic reference points. Therefore, for any

$\left[d_{ij}^{\vec{ab}}(t_k)^-, d_{ij}^{\vec{ab}}(t_k)^+\right]$, there is no gain or loss in the terms of personnel to the intelligent robot. The equation is as follows.

$$G_{ij}^{\vec{ab}}(t_k) = 0 \tag{19}$$

$$L_{ij}^{\vec{ab}}(t_k) = 0 \tag{20}$$

The sixth case involves the quantitative relationship $d_{ij}^{\vec{ab}}(t_k)^- < r^{\vec{ab}}(t_k)^- < r^{\vec{ab}}(t_k)^+ < d_{ij}^{\vec{ab}}(t_k)^+$. The interval number of dynamic reference points is a subset of the interval number of preference information. Therefore, it is necessary to divide the interval number of preference information into two parts: $\left[d_{ij}^{\vec{ab}}(t_k)^-, r^{\vec{ab}}(t_k)^-\right]$ and

$\left[r^{\vec{ab}}(t_k)^+, d_{ij}^{\vec{ab}}(t_k)^+\right]$. The impact of these parts on the return value and loss value can be determined from the corresponding calculation formulae involved in the third and fourth cases. They can be expressed as follows.

$$G_{ij}^{\vec{ab}}(t_k) = \int_{r^{\vec{ab}}(t_k)^+}^{d_{ij}^{\vec{ab}}(t_k)^+} \left(x - r^{\vec{ab}}(t_k)^+\right)f(x)dx = 0.5\left(d_{ij}^{\vec{ab}}(t_k)^+ - r^{\vec{ab}}(t_k)^+\right) \tag{21}$$

$$L_{ij}^{\vec{ab}}(t_k) = \int_{d_{ij}^{\vec{ab}}(t_k)^-}^{r^{\vec{ab}}(t_k)^-} \left(x - r^{\vec{ab}}(t_k)^-\right)f(x)dx = 0.5\left(d_{ij}^{\vec{ab}}(t_k)^- - r^{\vec{ab}}(t_k)^-\right) \tag{22}$$

The calculation formulae of income value and loss value under the aforementioned six cases are summarised in Table 3.

According to Table 3, the gain–loss matrix $GLM^{\vec{ab}}(t_k)$ of personnel to the intelligent robot in stage $t_k$ can be obtained. Similarly, the gain–loss matrix $GLM^{\vec{ac}}(t_k)$ of personnel to position and the gain–loss matrix $GLM^{\vec{bc}}(t_k)$ of the intelligent robot to position can be obtained.

Additionally, the gain–loss matrix of personnel to the intelligent robot is $GLM^{\vec{ab}}(t_k) = \left(\triangle x_{ij}^{\vec{ab}}(t_k)\right)_{m \times n}$. Here, $\triangle x_{ij}^{\vec{ab}}(t_k) = G_{ij}^{\vec{ab}}(t_k) + L_{ij}^{\vec{ab}}(t_k)$. Similarly, the gain–loss matrix of personnel to position is $GLM^{\vec{ac}}(t_k) = \left(\triangle x_{ij}^{\vec{ac}}(t_k)\right)_{m \times l}$. Here,

**Table 3. Calculation formula of benefits and losses of personnel to intelligent robot.**

| Case | Gains | Loss |
|------|-------|------|
| Case 1 | 0 | $0.5\left(d_{ij}^{\vec{ab}}(t_k)^- + d_{ij}^{\vec{ab}}(t_k)^+\right) - r^{\vec{ab}}(t_k)^-$ |
| Case 2 | $0.5\left(d_{ij}^{\vec{ab}}(t_k)^- + d_{ij}^{\vec{ab}}(t_k)^+\right) - r^{\vec{ab}}(t_k)^+$ | 0 |
| Case 3 | 0 | $0.5\left(d_{ij}^{\vec{ab}}(t_k)^- - r^{\vec{ab}}(t_k)^-\right)$ |
| Case 4 | $0.5\left(d_{ij}^{\vec{ab}}(t_k)^+ - r^{\vec{ab}}(t_k)^+\right)$ | 0 |
| Case 5 | 0 | 0 |
| Case 6 | $0.5\left(d_{ij}^{\vec{ab}}(t_k)^+ - r^{\vec{ab}}(t_k)^+\right)$ | $0.5\left(d_{ij}^{\vec{ab}}(t_k)^- - r^{\vec{ab}}(t_k)^-\right)$ |

$\triangle x_{ij}^{\vec{ac}}(t_k) = G_{ij}^{\vec{ac}}(t_k) + L_{ij}^{\vec{ac}}(t_k)$. The gain–loss matrix of the intelligent robot to position is $GLM^{\vec{bc}}(t_k) = \left(\triangle x_{ij}^{\vec{bc}}(t_k)\right)_{n \times l}$. Here, $\triangle x_{ij}^{\vec{bc}}(t_k) = G_{ij}^{\vec{bc}}(t_k) + L_{ij}^{\vec{bc}}(t_k)$.

## 3.4 Obtaining prospect value based on the value function

In the previous section, we described the transformation of preference information matrix into a gain–loss matrix. In this section, the prospect value matrix can be obtained according to prospect theory.

Prospect theory was first proposed by Kahneman and Tversky [35] in 1979, and it has been widely used in risk decision-making. The main contribution of this theory is that the decision-maker's psychological expectation factors are considered in the decision-making process. Furthermore, the decision-maker's preference is expressed by the decision-maker's subjective value feeling due to the decision-maker's psychological expected income and loss. This is similar to the income loss matrix obtained in the previous section. This value preference is termed as the prospect value in the prospect theory. It should be integrated through the value function and weight probability function. The value function [36] is defined as follows.

$$v(\triangle x) = \begin{cases} (\triangle x)^\alpha, & \triangle x \geq 0 \\ -\rho(-\triangle x)^\beta, & \triangle x < 0 \end{cases} \tag{23}$$

Here, $\triangle x$ denotes the deviation between the preference information and reference point, $\alpha$ and $\beta$ denote the sensitivity of the decision-maker to gains and losses, respectively, and $0 \leq \alpha \leq \beta \leq 1$. Furthermore, $\rho$ is the risk aversion coefficient. It is defined as the sensitivity of decision-makers to loss risk. $\rho > 1$ implies that the sensitivity of decision-makers to loss risk is higher than that to return risk. In the existing research, it is observed that the setting coefficient $\alpha = \beta = 0.88$ and $\rho = 2.25$ are consistent with the experience. The value function is generally illustrated as shown in Fig 2 for a better understanding.

In terms of gains, $\triangle x$ indicates the degree to which the preference information of the scheme deviates from the reference point. Its corresponding prospect value is $v(\triangle x)$. Similarly, in terms of loss, $-\triangle x$ indicates the degree of deviation of the preference information of the scheme from the reference point. Its corresponding prospect value is $v(-\triangle x)$. Therefore, according to Eq (21), the gain–loss matrix can be transformed into the prospect value matrix.

The prospect value matrices of personnel to the intelligent robot, personnel to position, and intelligent robot to position are expressed as $O^{\vec{ab}}(t_k) = \left(v\left(\triangle x_{ij}^{\vec{ab}}(t_k)\right)\right)_{m \times n}$,

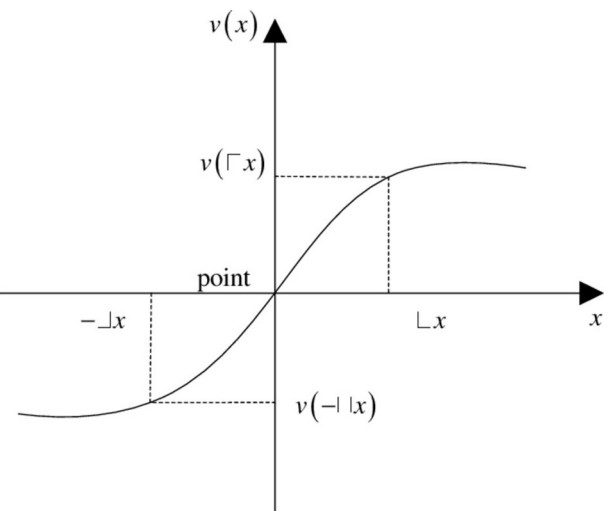

**Fig 2. Value function of prospect theory.**

$O^{\vec{ac}}(t_k) = \left( v\left( \triangle x_{ij}^{\vec{ac}}(t_k) \right) \right)_{m \times l}$, and $O^{\vec{bc}}(t_k) = \left( v\left( \triangle x_{ij}^{\vec{bc}}(t_k) \right) \right)_{n \times l}$, respectively. The next section transforms the prospect value matrix into a satisfaction matrix.

## 3.5 Calculation of satisfaction matrix based on attenuation index model

The completeness of the information is related to the proportion of weight in different stages. In the actual three-sided matching problem, the matching subject has different information in different stages. In general, closer to the end of decision-making, the information that the decision-maker possesses is more complete, and the impact on the matching result is higher. Conversely, in the early stage of decision-making, the information that the decision-maker possesses is sparser, and the impact on the final matching result is less. Therefore, in the process of dynamic matching, the preference information of different stages should be weighted. Several studies have determined that the variation trend of weight in different stages has the characteristics of exponential attenuation. Therefore, the exponential attenuation model is introduced to obtain the weight of each stage, and the satisfaction of each stage is weighted and integrated.

**Definition 5** [37] Let $T = \{t_1, t_2, \ldots t_p\}$ be different stages of the three-sided matching decision, and let weight of stage $t_k$ be $w_k$. Then,

$$w_k = C_0 e^{\lambda\left( t_k - t_p \right)} \tag{24}$$

where is the constant and is the attenuation coefficient.

The weight sum of each stage is one. Therefore, $\sum_{k=1}^{p} w(t_k) = \sum_{k=1}^{p} C_0 e^{\lambda\left( t_k - t_p \right)} = 1$ and $C_0 = \dfrac{1}{\sum_{k=1}^{p} e^{\lambda\left( t_k - t_p \right)}}$.

$$w(t_k) = \frac{e^{\lambda t_k}}{\sum_{k=1}^{p} e^{\lambda t_k}}, \ k = 1, 2, \cdots p \tag{25}$$

In the real dynamic matching process, the value of attenuation coefficient $\lambda(0 < \lambda < 1)$ has

important practical significance. It reflects the cumulative rate of information mastered by decision-makers over time. The matching decision-makers can determine $\lambda$ according to the field situation.

By weighting the prospect value of each stage,

$$\alpha_{ij} = \sum\nolimits_{k=1}^{P} O^{\vec{ab}}(t_k)w(t_k), \tag{26}$$

$$\beta_{ik} = \sum\nolimits_{k=1}^{P} O^{\vec{ac}}(t_k)w(t_k), \tag{27}$$

$$\gamma_{jk} = \sum\nolimits_{k=1}^{P} O^{\vec{bc}}(t_k)w(t_k). \tag{28}$$

The comprehensive satisfaction matrix $\alpha = [\alpha_{ij}]_{m \times n}$, $\beta = [\beta_{ik}]_{m \times l}$, and $\gamma = [\gamma_{jk}]_{n \times l}$ can be obtained.

## 4. Dynamic three-sided matching model of PRPM in intelligent assembly line

Considering the maximum satisfaction of the three sides as the objective, according to the aforementioned comprehensive satisfaction matrix $\alpha = [\alpha_{ij}]_{m \times n}$, $\beta = [\beta_{ik}]_{m \times l}$, and $\gamma = [\gamma_{jk}]_{n \times l}$ an optimal matching model is established and solved to obtain the matching results.

First, $x_{ijk}$ denotes a 0–1 variable.

$$x_{ijk} \begin{cases} 1, \left(a_i, b_j, c_k\right) \in M; \\ 0, \left(a_i, b_j, c_k\right) \notin M. \end{cases} \tag{29}$$

where $a_i$, $b_j$, $c_k$ denote a matching group $(a_i, b_j, c_k)$ in $M$ if $x_{ijk} = 1$, and if $x_{ijk} = 0$, then the matching group is not in $M$.

Additionally, there are three objective functions in the model. The unilateral matching satisfaction of personnel relative to the intelligent robot is as follows:

$$Z_A = \sum_{i=1}^{m} \sum_{j=1}^{n} \sum_{k=1}^{l} \alpha_{ij} x_{ijk} \tag{30}$$

Similarly, the unilateral matching satisfaction of personnel relative to position and unilateral matching satisfaction of the intelligent robot relative to position are as follows.

$$Z_B = \sum_{i=1}^{m} \sum_{j=1}^{n} \sum_{k=1}^{l} \beta_{ij} x_{ijk} \tag{31}$$

$$Z_C = \sum_{i=1}^{m} \sum_{j=1}^{n} \sum_{k=1}^{l} \gamma_{ij} x_{ijk} \tag{32}$$

Therefore, the following multi-objective three-sided matching model is constructed by considering the maximum satisfaction of unilateral matching of the three parties as the objective and complete matching of side $C$ (ensuring the complete matching of production position) as

the constraint.

$$MaxZ_A = \sum_{i=1}^{m} \sum_{j=1}^{n} \sum_{k=1}^{l} \alpha_{ij} x_{ijk} \tag{33}$$

$$MaxZ_B = \sum_{i=1}^{m} \sum_{j=1}^{n} \sum_{k=1}^{l} \beta_{ij} x_{ijk} \tag{34}$$

$$MaxZ_C = \sum_{i=1}^{m} \sum_{j=1}^{n} \sum_{k=1}^{l} \gamma_{ij} x_{ijk} \tag{35}$$

$$s.t. \ \sum_{j=1}^{n} \sum_{k=1}^{l} x_{ijk} \leq 1, i = 1, \cdots, m \tag{36}$$

$$\sum_{i=1}^{m} \sum_{i=1}^{n} x_{ijk} \leq 1, j = 1, \cdots, n \tag{37}$$

$$\sum_{i=1}^{m} \sum_{j=1}^{n} x_{ijk} = 1, k = 1, \cdots, l \tag{38}$$

$$x_{ijk} = 0 \ or \ 1 \tag{39}$$

## 5. Solution strategy of the model

The aforementioned model belongs to a multi-objective 0–1 integer programming problem. To solve it conveniently, the linear weighting method is used to transform the multi-objective model into a single objective model for optimisation. The weight of each party is expressed in degrees using the triangle balance principle. The proportion of personnel, proportion of intelligent robot, and proportion of position can be expressed as $T_a + T_b + T_c = 180°$.

The weight ratio of the three sides can be expressed as $\eta_1 = \frac{T_a}{180°}$, $\eta_2 = \frac{T_b}{180°}$, and $\eta_3 = \frac{T_c}{180°}$. Therefore, the objective function of three-sided matching can be transformed into $Z = \eta_1 Z_A + \eta_2 Z_B + \eta_3 Z_C$.

$$MaxZ = \eta_1 Z_A + \eta_2 Z_B + \eta_3 Z_C \tag{40}$$

$$s.t. \sum_{j=1}^{n} \sum_{k=1}^{l} x_{ijk} \leq 1, i = 1, \cdots, m \tag{41}$$

$$\sum_{i=1}^{m} \sum_{k=1}^{l} x_{ijk} \leq 1, j = 1, \cdots, n \tag{42}$$

$$\sum_{i=1}^{m} \sum_{j=1}^{n} x_{ijk} = 1, k = 1, \cdots, l \tag{43}$$

$$x_{ijk} = 0 \ or \ 1 \tag{44}$$

Model (40)–(44) is a classical model for assignment problems. Simultaneously, the

proposed problem is an $n \times n \times n$ assignment problem. Certain polynomial time algorithms have been proposed to solve the classical assignment problems, such as the Kuhn–Munkres algorithm [38] and Bertsekas algorithm [39]. Furthermore, the existing assignment algorithms can be used to conveniently solve models (18)–(22). Furthermore, the proposed model can also be solved by applying mathematical optimisation software (e.g. LINGO 11.0).

As mentioned above, the matching algorithm is designed to address the proposed model. Specifically, the pseudocodes of this proposed two-sided matching model can be summarised as follows:

Input: $\alpha = [\alpha_{ij}]_{m \times n}, \beta = [\beta_{ik}]_{m \times l}, \gamma = [\gamma_{jk}]_{n \times l}$ THE MATCHING MATRIX
Output: $X = [x_{ijk}]_{m \times n \times l}, Z_{max}$ THE MAXIMUM SATISFACTION

```
1: MODEL:

2: SETS:

3: A/1...m/;

4: B/1...n/;

5: C/1...l/;

6: LINK (A, B C):α, β, γ;

7: END SET

8: INPUTDATA: THE MATCHING MATRIX:

9: α = [αij]m × n, β = [βik]m × l, γ = [γjk]n × l;

10: END DATA

11: MAX = @SUM(LINK: α*X+β*X+γ*X);

12: @FOR(A(I):@SUM(A(I):X(I,J,K))< = 1);

13: @FOR(B(J):@SUM(B(J):X(I,J,K))< = 1);

14: @FOR(B(J):@SUM(C(L):X(I,J,K)) = 1);

15: @FOR (LINK:@BIN(X));

16: OUTPUT X = [xijk]m × n × l, Zmax THE MAXIMUM SATISFACTION

17: END
```

Therefore, the multi-objective model is transformed into a single objective model with the maximum satisfaction of three-sided weighted matching as the objective. The solution steps of the three-sided dynamic matching model are shown in Fig 3.

As shown in Fig 3, the steps of three-sided dynamic matching are as follows:

Step 1: The multi-level multi-stage evaluation information is obtained based on the uncertain fuzzy language set.

Step 2: The initial reference point is obtained based on Eqs (1)–(3), and the overall preference expectation is obtained based on Eqs (4–6). Then, the dynamic reference point is obtained based on Eqs (7)–(9).

Step 3: Based on the probability density function, the gain–loss matrix is calculated using the formulae of six cases shown in Table 3.

Step 4: The prospect value matrix is calculated based on Eq (22).

Step 5: To obtain the weight value of each stage based on Eq (23), the comprehensive satisfaction matrix of the three-sided subject are calculated based on Eqs (24)–(26), and the multi-objective three-sided matching model (Eqs (31)–(37)) is constructed.

Step 6: The triangular balance principle is used to convert the multi-target three-sided matching into the single-target matching model.

Step 7: Data analysis software LINGO 11 is used to solve it. Finally, the optimal matching scheme is obtained.

## 6. Case study

During the construction of the production line, it is expected that the operators will include mature employees and newly recruited employees from other production lines, and the robots will include newly purchased robots of various models and old robots retained for transformation. Therefore, the constructors of production lines need to evaluate the operators and robots, and identify the key stations, to put high-quality and suitable robots and operators into the key stations.

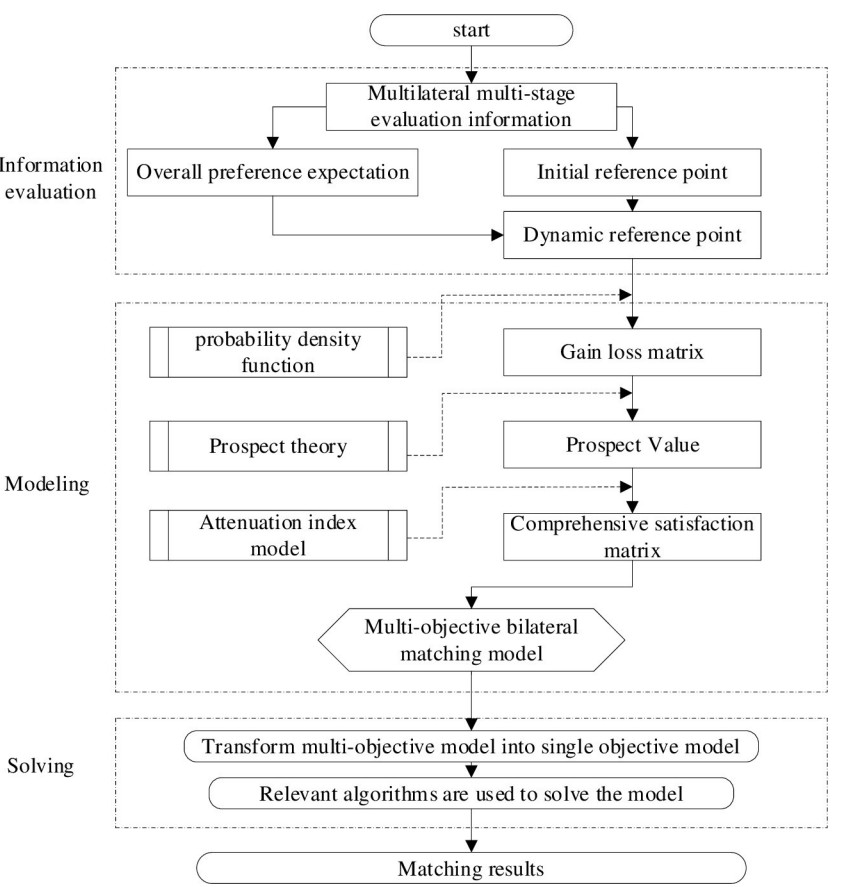

**Fig 3. Solution steps of three-sided dynamic matching.**

Currently, a manufacturing plant should transform and upgrade four key positions $C = \{C_1, C_2, C_3, C_4\}$. $C_1$ position completes part of the assembly tasks in the rear floor area of the car body. The work content of this station includes high-strength structural parts of the car body, and its dimensional accuracy and stability are poor. $C_2$ position and $C_3$ position are a pair of left and right mirror stations, which complete part of the assembly tasks on the right and left side of the body side wall. The work involves many parts, and has a great impact on the assembly of left and right doors, cockpit and other performance. $C_4$ position is $C_2$ position next to the previous position, and there are relatively few assembly parts. In structure, it is next to the operating parts of $C_1$ position, so the operation content of this position has a great impact on the function.

The production tasks corresponding to each position are completed by intelligent robots and personnel. To reduce the cooperation risks among positions, intelligent robots, and personnel, and improve the benefits and matching efficiency of the three parties, the actual operation status of five intelligent robots $B = \{B_1, B_2, B_3, B_4, B_5\}$ and six employees $A = \{A_1, A_2, A_3, A_4, A_5, A_6\}$ in the previous nine months and the working conditions of employees in the manufacturing plant are investigated. Each stage is considered to comprise three months. Additionally, the manufacturing plant invites many experts (process engineers, human resource managers, and mechanical maintenance personnel) to evaluate the performance of the three sides in recent months for reasonable matching results. In our study, preference information used in the experiment is acquired from an automobile manufacturer in Chongqing. Furthermore, the information recording centre collects, classifies, pre-processes, and integrates various matching subjects based on different stages of manufacturing. These statistics mainly reflect the positions, intelligent robots, and employees over the entire manufacturing life cycle.

## 6.1 Dynamic three-sided matching results of PRPM

In terms of personnel, it is necessary to comprehensively consider the safety of personnel operating machinery, operation proficiency, salary, physical condition, and similar position experience of intelligent robots. In terms of intelligent robots, it is necessary to comprehensively consider the production process, warranty degree, service life, failure rate, and energy consumption. In terms of position, it is necessary to comprehensively consider the quality requirements, skill level, and position attributes of the production tasks corresponding to the position. Based on the aforementioned factors, experts use the fuzzy linguistic preference information (Table 1) to evaluate the three sides. The evaluation results are presented in the form of interval numbers as listed in Tables A.1–A.3 (See S1 File).

The initial reference points of each stage are obtained according to Eqs (1)–(3) as listed in Table 4.

The overall expectations are obtained based on Eqs (4)–(6). The overall expectation values of personnel to intelligent robot, personnel to position, and intelligent robot to position are [0.502,0.702], [0.456,0.656], and [0.453,0.653], respectively.

Then, dynamic reference points for each stage are calculated (see Table 5).

Based on the probability density function, the dynamic reference point and initial evaluation matrix are substituted into the formulae of six cases in Table 3 to obtain the gain–loss matrix of each stage. Then, prospect theory is used to convert the gain–loss matrix into the prospect value matrix. Thus, the prospect value matrix of each stage is obtained based on Eq (23) as listed in Tables A.4–A.6 (see S1 File).

The next step involves calculating the weight value of each stage. The attenuation index model is introduced to convert the prospect value matrix into the satisfaction matrix.

**Table 4. Initial reference points of each stage.**

|  | $T_1$ | $T_2$ | $T_3$ |
|---|---|---|---|
| Personnel to intelligent robot | [0.487,0.687] | [0.52,0.72] | [0.5,0.7] |
| Personnel to position | [0.458,0.658] | [0.408,0.608] | [0.5,0.7] |
| Intelligent robot to position | [0.45,0.65] | [0.52,0.72] | [0.39,0.59] |

Therefore, the weight values of the three stages are obtained based on formula 6.25, and the weight values of the first, second, and third stages are 0.186, 0.307, and 0.507, respectively. Then, according to Eqs (26)–(28), the multi-stage prospect value matrix is transformed into the final satisfaction matrix as follows: $\alpha = [\alpha_{ij}]_{m \times n}$, $\beta = [\beta_{ik}]_{m \times b}$, and $\gamma = [\gamma_{jk}]_{n \times l}$

$$\alpha = \begin{bmatrix} 0.259 & -0.431 & 0.260 & 0.141 & 0.259 \\ -0.078 & -0.357 & 0.023 & 0.066 & 0.174 \\ 0.153 & -0.580 & -0.550 & -0.369 & 0.184 \\ 0.207 & -0.174 & -0.583 & -0.168 & -0.266 \\ 0.043 & -0.053 & -0.406 & 0.023 & 0.072 \\ 0.239 & -0.742 & 0.072 & 0.331 & -0.248 \end{bmatrix}$$

$$\beta = \begin{bmatrix} -0.671 & -0.264 & -0.200 & -0.411 \\ -0.274 & -0.334 & -0.030 & -0.309 \\ 0.049 & -0.166 & -0.073 & 0.007 \\ 0.121 & 0.149 & -0.109 & -0.073 \\ -0.237 & 0.121 & 0.000 & -0.030 \\ 0.037 & 0.213 & -0.171 & -0.484 \end{bmatrix}$$

$$\gamma = \begin{bmatrix} 0.199 & -0.054 & 0.122 & 0.055 \\ 0.095 & 0.139 & -0.239 & 0.103 \\ 0.095 & -0.296 & 0.313 & -0.150 \\ -0.377 & 0.220 & 0.239 & 0.141 \\ -0.799 & -0.317 & -0.745 & -0.338 \end{bmatrix}$$

Finally, let $T_a = 90°$, $T_b = 45°$, and $T_c = 45°$. It is assumed that decision-makers prefer personnel and that the proportions of intelligent robots and positions are equal. The satisfaction matrix is substituted into a three-sided matching model, and the final matching scheme is calculated using LINGO 11 software. The optimal value is 0.624, and the optimal matching solution is $x_{133} = x_{354} = x_{411} = x_{642} = 1$. Personnel $A_1$ and intelligent robot $B_3$ are matched to the position $C_3$, personnel $A_3$ and intelligent robot $B_5$ are matched to the position $C_4$, personnel $A_4$ and intelligent robot $B_1$ are matched to the position $C_1$, and personnel $A_6$ and intelligent robot $B_4$ are matched to the position $C_2$. Among these, personnel $A_2$, $A_5$ do not match with any intelligent robot or position, intelligent robot $B_2$ does not match, and all positions are matched. This shows the effectiveness of the proposed method.

**Table 5. Dynamic reference points of each stage.**

|  | $T_1$ | $T_2$ | $T_3$ |
|---|---|---|---|
| Personnel to intelligent robot | [0.494,0.694] | [0.511,0.711] | [0.501,0.701] |
| Personnel to position | [0.457,0.657] | [0.432,0.632] | [0.478,0.678] |
| Intelligent robot to position | [0.412,0.652] | [0.487,0.687] | [0.422,0.622] |

## 6.2 Discussion

In this section, further discussions are presented to demonstrate the characteristics of the proposed method. The proposed method exhibits the following characteristics when compared with previous methods.

First, the matching results of the model are observed by adjusting the weights of the three objective functions. In this case, the designed weights are 0.5, 0.25, and 0.25, respectively. Therefore, we redesign different weight coefficients to observe the calculation results (see Table 6).

As shown in the table above, different weight values of the objective function can affect the final matching scheme. The main positions can be matched with different personnel and intelligent robots. When the weight of the position is high, the position can be matched with personnel and intelligent robots. When the weight of the personnel is high, the position can be matched with personnel and intelligent robots.

Second, the influence of the attenuation coefficient on matching is analysed. The attenuation coefficient is the key parameter affecting the final satisfaction matrix. It is necessary to further examine the results of dynamic trilateral matching under different attenuation coefficients.

Therefore, in the case wherein other parameters do not vary, the attenuation coefficient should be adjusted. The dynamic three-sided matching results under five attenuation conditions should be set as $\lambda = 0.1, 0.3, 0.5, 0.7, 0.9$. First, the fluctuation diagram of weight in each stage under different attenuation coefficients should be calculated (see Fig 4).

As shown in Fig 4, the higher the attenuation coefficient, the higher the weight of the third stage. In contrast, the weight of the first stage decreases gradually. In line with the definition of

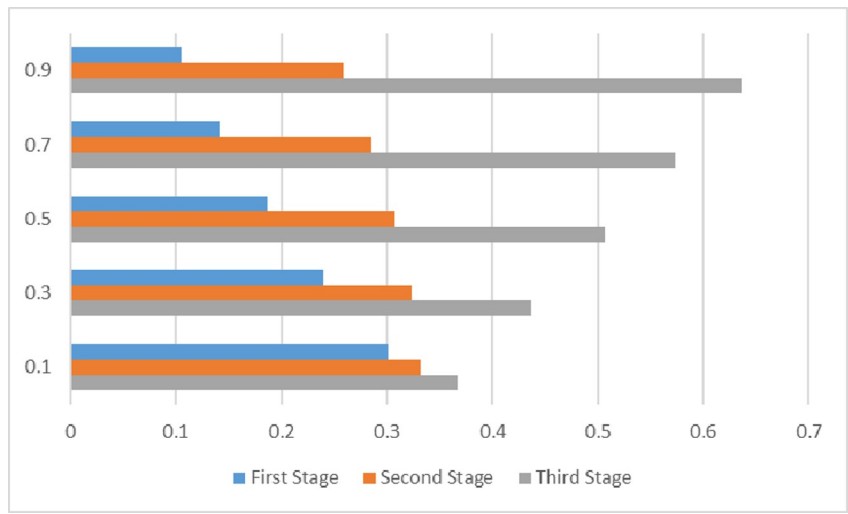

**Fig 4. Influence of attenuation coefficient on weight of each stage.**

**Table 6. Parameter analysis of the weight of the objective function.**

| Weight coefficient of objective function | Matching scheme | Objective function value |
|---|---|---|
| 0.8, 0.1, 0.1 | $x_{133} = x_{354} = x_{411} = x_{642} = 1$ | 0.8391 |
| 0.6, 0.2, 0.2 | $x_{133} = x_{354} = x_{411} = x_{642} = 1$ | 0.6962 |
| 0.4, 0.3, 0.3 | $x_{133} = x_{524} = x_{411} = x_{642} = 1$ | 0.5797 |
| 0.2, 0.4, 0.4 | $x_{233} = x_{524} = x_{411} = x_{642} = 1$ | 0.5452 |
| 0.1, 0.1, 0.8 | $x_{133} = x_{524} = x_{411} = x_{642} = 1$ | 0.7529 |

the attenuation model, the degree of information attenuation is also higher. This implies that the impact of the initial evaluation information on the final decision is weaker. The final three-sided matching schemes under different attenuation coefficients and the corresponding results are listed in Table 7.

As shown in this table, the increase in attenuation degree mainly affects two positions $C_1$, $C_4$ wherein the matching objects in terms of personnel and intelligent robot vary, and the value of the objective function increases progressively. Therefore, in the final decision-making process, the decision-maker should determine the attenuation coefficient according to the actual variation in production. If the variation in production is high, the proportion of evaluation information in the previous stage should be marginal, and the attenuation coefficient should select a larger parameter. The attenuation coefficient should select a smaller parameter if the production condition is relatively stable.

Third, the proposed method can address the dynamic three-sided matching problem by considering multiple stages and the psychological behaviour of decision-makers. In previous studies, different approaches have been used to solve the three-sided matching problem [40, 41]. However, it is generally assumed that this problem is static. To the best of our knowledge, the findings by Yang [21] can be used to address preference information by using the preference order of this problem. However, Yang [21] focused on the approach for addressing the static three-sided matching problem, which did not consider the psychological behaviour of decision-makers. Conversely, the proposed method transcends the boundary of three-sided matching research. It represents a new solution for multi-stage dynamic matching and considers the decision psychological preference of decision-makers when they master different levels of information in different stages. Therefore, the proposed method can help obtain new insights on the dynamic three-sided matching problem.

## 7. Conclusions

This paper dynamic three-sided matching model aimed at the three-sided matching decision-making problem with multi-stage preference information in an intelligent environment. First, the problem of man–machine–position dynamic three-sided matching in an intelligent assembly line is proposed. Second, uncertain language is used to obtain the preference information

**Table 7. Parameter analysis of objective function.**

| Attenuation coefficient | Matching scheme | Objective function value |
|---|---|---|
| 0.1 | $x_{133} = x_{314} = x_{421} = x_{642} = 1$ | 0.494 |
| 0.3 | $x_{133} = x_{354} = x_{411} = x_{642} = 1$ | 0.558 |
| 0.5 | $x_{133} = x_{354} = x_{411} = x_{642} = 1$ | 0.625 |
| 0.7 | $x_{133} = x_{354} = x_{411} = x_{642} = 1$ | 0.687 |
| 0.9 | $x_{133} = x_{411} = x_{554} = x_{642} = 1$ | 0.753 |

of multi-stage matching subjects. Third, the preference information is integrated with the prospect theory to obtain the development trend of the same matching subject in different stages, namely, the setting of dynamic reference points, calculation of income loss matrix based on the probability density function, and determination of the prospect value matrix based on the value function. Then, based on the attenuation index model, the prospect value matrix is transformed into a satisfaction matrix. The results show that the establishment of a multi-objective model can solve the optimal matching scheme of personnel–robot position dynamic three-sided matching in an intelligent assembly line. The proposed method can objectively and effectively reflect the variation trend of matching subjects in multiple stages.

However, the proposed method has certain limitations. Multi-granular HFLTSs are generally elicited by decision-makers in a multi-attribute group decision-making (MAGDM) problem owing to the uncertainty of the decision environment as well as differences in decision-makers' culture and background knowledge [42]. Therefore, in future studies, multi-granularity HFLTSs should be incorporated into the dynamic three-sided matching problem.

## Supporting information

**S1 File.**
(DOCX)

## Author Contributions

**Conceptualization:** Zhi-chao Liang.

**Funding acquisition:** Yu Yang, Qiu Xie, Bao-dong Li.

**Investigation:** Xue-jiao Zhang.

**Methodology:** Zhi-chao Liang, Yu Yang, Qiu Xie, Xue-jiao Zhang, Bao-dong Li.

**Resources:** Zhi-chao Liang, Jing Wang.

**Software:** Jing Wang.

**Supervision:** Yu Yang.

**Writing – original draft:** Zhi-chao Liang.

**Writing – review & editing:** Yu Yang, Qiu Xie.

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
