## [Decision Letter · Decision Letter 0]

27 Jun 2022

PONE-D-22-06663Dynamic three-sided matching model for PRPM in intelligent environmentPLOS ONE

Dear Dr. XIE,

Thank you for submitting your manuscript to PLOS ONE. After careful consideration, we feel that it has merit but does not fully meet PLOS ONE’s publication criteria as it currently stands. Therefore, we invite you to submit a revised version of the manuscript that addresses the points raised during the review process.

We look forward to receiving your revised manuscript.

Kind regards,

Ziqiang Zeng, Ph.D.

Academic Editor

PLOS ONE

Journal Requirements:

This study was supported by Fundamental Research Funds for the Central Universities-China (No. 2020CDJSK03PT08, 2021CDJSKJC17 and 2020CDCGJX029), National Natural Science Foundation of China (Grant No. 71571023) and the Scientific and Technological Research Program of Chongqing Municipal Education Commission (Grant No. KJQN202101617).

This study was supported by Fundamental Research Funds for the Central Universities-China (No. 2020CDJSK03PT08, 2021CDJSKJC17 and 2020CDCGJX029), National Natural Science Foundation of China (Grant No. 71571023) and the Scientific and Technological Research Program of Chongqing Municipal Education Commission (Grant No. KJQN202101617).

This study was supported by Fundamental Research Funds for the Central Universities-China (No. 2020CDJSK03PT08, 2021CDJSKJC17 and 2020CDCGJX029), National Natural Science Foundation of China (Grant No. 71571023) and the Scientific and Technological Research Program of Chongqing Municipal Education Commission (Grant No. KJQN202101617).

The authors declare that they have no known competing financial interests or personal relationships that could have appeared to influence the work reported in this paper.

6. Please upload a copy of Figure 6, to which you refer in your text on page 15. If the figure is no longer to be included as part of the submission please remove all reference to it within the text.

Reviewers' comments:

Reviewer's Responses to Questions

**Comments to the Author**

1. Is the manuscript technically sound, and do the data support the conclusions?

Reviewer #1: Yes

Reviewer #2: Yes

2. Has the statistical analysis been performed appropriately and rigorously? 

Reviewer #1: N/A

Reviewer #2: Yes

3. Have the authors made all data underlying the findings in their manuscript fully available?

Reviewer #1: Yes

Reviewer #2: No

4. Is the manuscript presented in an intelligible fashion and written in standard English?

Reviewer #1: No

Reviewer #2: Yes

5. Review Comments to the Author

Reviewer #1: This paper proposed a dynamic three-sided decision model and applied it in a personnel–robot-position matching (PRPM) problem, which is an interesting topic. However, this paper should be revised carefully. My comments are attached.

Reviewer #2: 1. This paper designs a dynamic three-sided matching decision-making approach focusing on the three sides matching problem. I am confused that whether this problem is NP-hardness? If yes, whether the proposed algorithm can be completed within polynomial time? Then, what’s the approximate ratio of the output result?

2. There are lots of methods based on dynamic matching theory introduced in the introduction of this paper, so I am curious that what’s the biggest improvement from your proposition over these works? Is it just a simple generalization from these work to your work?

3. More importantly, the contributions of this work are not very clear, so I think authors need to highlight their innovations in the introduction section.

4. In the simulation, whether an algorithm is needed as a benchmark to be compared with your algorithm? And a recurrent learning process is required to enhance the accuracy of the output? If not, how to guarantee the matching accuracy of your algorithm?

6. PLOS authors have the option to publish the peer review history of their article (what does this mean?). If published, this will include your full peer review and any attached files.

Reviewer #1: No

Reviewer #2: No

---

## [Author Response · Author response to Decision Letter 0]

11 Aug 2022

We thank the reviewers for their suggestions and help to improve the level of the manuscript. See the attachment for specific modification instructions. The new manuscript has many modifications, so the annotation mode is not adopted.

---

## [Decision Letter · Decision Letter 1]

4 Sep 2022

PONE-D-22-06663R1Dynamic three-sided matching model for personnel–robot-position matching problem in intelligent environmentPLOS ONE

Dear Dr. XIE,

Thank you for submitting your manuscript to PLOS ONE. After careful consideration, we feel that it has merit but does not fully meet PLOS ONE’s publication criteria as it currently stands. Therefore, we invite you to submit a revised version of the manuscript that addresses the points raised during the review process.

We look forward to receiving your revised manuscript.

Kind regards,

Ziqiang Zeng, Ph.D.

Academic Editor

PLOS ONE

Journal Requirements:

Reviewers' comments:

Reviewer's Responses to Questions

**Comments to the Author**

1. If the authors have adequately addressed your comments raised in a previous round of review and you feel that this manuscript is now acceptable for publication, you may indicate that here to bypass the “Comments to the Author” section, enter your conflict of interest statement in the “Confidential to Editor” section, and submit your "Accept" recommendation.

Reviewer #1: (No Response)

Reviewer #2: All comments have been addressed

2. Is the manuscript technically sound, and do the data support the conclusions?

Reviewer #1: Partly

Reviewer #2: Yes

3. Has the statistical analysis been performed appropriately and rigorously? 

Reviewer #1: N/A

Reviewer #2: Yes

4. Have the authors made all data underlying the findings in their manuscript fully available?

Reviewer #1: No

Reviewer #2: Yes

5. Is the manuscript presented in an intelligible fashion and written in standard English?

Reviewer #1: No

Reviewer #2: Yes

6. Review Comments to the Author

Reviewer #1: Although some revisions have been made, there are still many problems that prevent the acceptcance of this paper. The authors should further revise the paper based on the following comments:

1) The reference list are not updated. I still find many papers cited in the paper are quite old and not international journal papers. The authors do not provide a good response to my previous comment #6. Please carefully revise the reference list by following my previous comments.

2) Again, my previous comment #4 is not well addressed. There are many problems for the format of the reference list. The author’s name are not well displayed.

3) If you cannot provide numerical comparative studies, you can provide some discussions through comparing with exsiting studies. Comprarative studies are missing.

4) The language is still quite poor. Please ask a copyeditor to improve the language and a certification is needed.

Reviewer #2: The demonstration of the pseudo-codes of the proposed algorithm is not very standard, so please revise it in a more standard format. Authors have solved all my given comments, so this paper can be accepted for publication in this journal.

7. PLOS authors have the option to publish the peer review history of their article (what does this mean?). If published, this will include your full peer review and any attached files.

Reviewer #1: No

Reviewer #2: No

---

## [Author Response · Author response to Decision Letter 1]

22 Oct 2022

Thanks for the comments of the review experts to continue to help us improve the level of the paper. See the annex for the details of the revised reply.

---

## [Decision Letter · Decision Letter 2]

28 Nov 2022

PONE-D-22-06663R2Dynamic three-sided matching model for personnel–robot-position matching problem in intelligent environmentPLOS ONE

Dear Dr. XIE,

Thank you for submitting your manuscript to PLOS ONE. After careful consideration, we feel that it has merit but does not fully meet PLOS ONE’s publication criteria as it currently stands. Therefore, we invite you to submit a revised version of the manuscript that addresses the points raised during the review process.

We look forward to receiving your revised manuscript.

Kind regards,

Ziqiang Zeng, Ph.D.

Academic Editor

PLOS ONE

Journal Requirements:

Reviewers' comments:

Reviewer's Responses to Questions

**Comments to the Author**

1. If the authors have adequately addressed your comments raised in a previous round of review and you feel that this manuscript is now acceptable for publication, you may indicate that here to bypass the “Comments to the Author” section, enter your conflict of interest statement in the “Confidential to Editor” section, and submit your "Accept" recommendation.

Reviewer #1: All comments have been addressed

Reviewer #2: (No Response)

Reviewer #3: All comments have been addressed

Reviewer #4: All comments have been addressed

2. Is the manuscript technically sound, and do the data support the conclusions?

Reviewer #1: Yes

Reviewer #2: Yes

Reviewer #3: Yes

Reviewer #4: Yes

3. Has the statistical analysis been performed appropriately and rigorously? 

Reviewer #1: N/A

Reviewer #2: Yes

Reviewer #3: Yes

Reviewer #4: N/A

4. Have the authors made all data underlying the findings in their manuscript fully available?

Reviewer #1: Yes

Reviewer #2: Yes

Reviewer #3: Yes

Reviewer #4: Yes

5. Is the manuscript presented in an intelligible fashion and written in standard English?

Reviewer #1: Yes

Reviewer #2: Yes

Reviewer #3: Yes

Reviewer #4: Yes

6. Review Comments to the Author

Reviewer #1: I checked the manuscript and found that the paper has been well revised. As a result, the paper can be considered for publication.

Reviewer #2: This paper establishes a dynamic three-sided matching model striving to maximize the matching satisfaction of multiple sides. My reviewing comments are listed as follows:

1. English presentation needs to be improved significantly.

2. The layout of the paper is not particularly graceful, as the demonstration of tables, figures and formulas is not very standard.

3. It's not clear that how to use the fuzzy language to quantify the uncertain information characteristics, and whether to apply a fuzzy function and the if-then rule. Besides, some references focusing on the fuzzy method in intelligent environment are not included in the paper, such as

K. Liu et al., "Big Medical Data Decision-Making Intelligent System Exploiting Fuzzy Inference Logic for Prostate Cancer in Developing Countries," in IEEE Access, vol. 7, pp. 2348-2363, 2019, doi: 10.1109/ACCESS.2018.2886198.

4. The representativeness of the given case study is missing. Please add it.

5. The pseudo-code presentation is not very standard, especially lacking the inputs and outputs.

Reviewer #3: The authors accepted the comments, I recommend the paper to be published.

The authors accepted the comments, I recommend the paper to be published.

Reviewer #4: The authors considered all comments and revised manuscript well. It seems manuscript has a god potential for further research specially on mixed model assembly lines.

7. PLOS authors have the option to publish the peer review history of their article (what does this mean?). If published, this will include your full peer review and any attached files.

Reviewer #1: No

Reviewer #2: No

Reviewer #3: No

Reviewer #4: No

---

## [Author Response · Author response to Decision Letter 2]

6 Jan 2023

Reviewer #2: This paper establishes a dynamic three-sided matching model striving to maximize the matching satisfaction of multiple sides. My reviewing comments are listed as follows:

1. English presentation needs to be improved significantly.

Response: The manuscripts were rechecked and proofread for language, grammar, phrase, clarity, and readability with the professional help of the Elsevier Institute. 

2. The layout of the paper is not particularly graceful, as the demonstration of tables, figures and formulas is not very standard.

Response: Revise charts, formulas, etc. in strict accordance with the requirements. Figure 4 shows the results of the LINGO software calculation model. After the consensus of the authors, it is unnecessary to put the screenshot of the LINGO software calculation results in the text, because there is no more important information except the maximum objective function value.

3. It's not clear that how to use the fuzzy language to quantify the uncertain information characteristics, and whether to apply a fuzzy function and the if-then rule. Besides, some references focusing on the fuzzy method in intelligent environment are not included in the paper, such as

K. Liu et al., "Big Medical Data Decision-Making Intelligent System Exploiting Fuzzy Inference Logic for Prostate Cancer in Developing Countries," in IEEE Access, vol. 7, pp. 2348-2363, 2019, doi: 10.1109/ACCESS.2018.2886198 Add to Citavi project by DOI.

Response: The advantage of the fuzzy language used in the manuscript is that experts can quickly give the matching degree between matching subjects according to experience. See Section 3.1 for details. Secondly, the literature Big Medical Data Decision Making Intelligent System Exploring Fuzzy Input Logic forState Cancer in Developing Countries has been added in the third paragraph of Section 1: Obvious, fuzzy sets have been widely used in the three sided matching problem. The fuzzy set theory has also been successfully applied in the Big medical data decision-making intelligent system to observe the curative powers of the deterministic treatment method on patients in real time[27].

4. The representativeness of the given case study is missing. Please add it.

Response: SECTION6: During the construction of the production line, it is expected that the operators will include mature employees and newly recruited employees from other production lines, and the robots will include newly purchased robots of various models and old robots retained for transformation. Therefore, the constructors of production lines need to evaluate the operators and robots, and identify the key stations, to put high-quality and suitable robots and operators into the key stations.

C1 position completes part of the assembly tasks in the rear floor area of the car body. The work content of this station includes high-strength structural parts of the car body, and its dimensional accuracy and stability are poor. C2 position and C3 position are a pair of left and right mirror stations, which complete part of the assembly tasks on the right and left side of the body side wall. The work involves many parts, and has a great impact on the assembly of left and right doors, cockpit and other performance. C4 position is C2 position next to the previous position, and there are relatively few assembly parts. In structure, it is next to the operating parts of C1 position, so the operation content of this position has a great impact on the function.

In our study, preference information used in the experiment is acquired from an automobile manufacturer in Chongqing. Furthermore, the information recording centre collects, classifies, pre-processes, and integrates various matching subjects based on different stages of manufacturing. These statistics mainly reflect the positions, intelligent robots, and employees over the entire manufacturing life cycle.

5. The pseudo-code presentation is not very standard, especially lacking the inputs and outputs.

Response: The writing method of pseudo code in page 11 of reference [27], and modify the code as required.

---

## [Decision Letter · Decision Letter 3]

14 Feb 2023

Dynamic three-sided matching model for personnel–robot-position matching problem in intelligent environments

PONE-D-22-06663R3

Dear Dr. XIE,

We’re pleased to inform you that your manuscript has been judged scientifically suitable for publication and will be formally accepted for publication once it meets all outstanding technical requirements.

Kind regards,

Ziqiang Zeng, Ph.D.

Academic Editor

PLOS ONE

Additional Editor Comments (optional):

Reviewers' comments:

Reviewer's Responses to Questions

**Comments to the Author**

1. If the authors have adequately addressed your comments raised in a previous round of review and you feel that this manuscript is now acceptable for publication, you may indicate that here to bypass the “Comments to the Author” section, enter your conflict of interest statement in the “Confidential to Editor” section, and submit your "Accept" recommendation.

Reviewer #1: All comments have been addressed

Reviewer #2: All comments have been addressed

Reviewer #3: All comments have been addressed

Reviewer #4: All comments have been addressed

2. Is the manuscript technically sound, and do the data support the conclusions?

Reviewer #1: Yes

Reviewer #2: (No Response)

Reviewer #3: Yes

Reviewer #4: Yes

3. Has the statistical analysis been performed appropriately and rigorously? 

Reviewer #1: N/A

Reviewer #2: (No Response)

Reviewer #3: Yes

Reviewer #4: N/A

4. Have the authors made all data underlying the findings in their manuscript fully available?

Reviewer #1: Yes

Reviewer #2: (No Response)

Reviewer #3: Yes

Reviewer #4: Yes

5. Is the manuscript presented in an intelligible fashion and written in standard English?

Reviewer #1: Yes

Reviewer #2: (No Response)

Reviewer #3: Yes

Reviewer #4: Yes

6. Review Comments to the Author

Reviewer #1: The authors have made necessary revisions to improve the quality of the paper and it can be considered for publication. Please check the format of the reference when submitting the final files.

Reviewer #2: (No Response)

Reviewer #3: The authors accepted the comments, I recommend the paper to be published, thanks. The authors accepted the comments, I recommend the paper to be published, thanks.

Reviewer #4: Authors proposed Dynamic three-sided matching model for personnel–robot-position matching problem in

intelligent environments.

All comments has been considered well.

7. PLOS authors have the option to publish the peer review history of their article (what does this mean?). If published, this will include your full peer review and any attached files.

Reviewer #1: No

Reviewer #2: No

Reviewer #3: No

Reviewer #4: No

---

## [Editor Report · Acceptance letter]

27 Feb 2023

PONE-D-22-06663R3 

Dynamic three-sided matching model for personnel–robot-position matching problem in intelligent environments 

Dear Dr. Xie:

I'm pleased to inform you that your manuscript has been deemed suitable for publication in PLOS ONE. Congratulations! Your manuscript is now with our production department. 

Kind regards, 

on behalf of

Dr. Ziqiang Zeng 

Academic Editor

PLOS ONE